# Preeclampsia and Cardiovascular Risk for Offspring

**DOI:** 10.3390/jcm10143154

**Published:** 2021-07-16

**Authors:** Wiktor Wojczakowski, Żaneta Kimber-Trojnar, Filip Dziwisz, Magdalena Słodzińska, Hubert Słodziński, Bożena Leszczyńska-Gorzelak

**Affiliations:** 1Department of Obstetrics and Perinatology, Medical University of Lublin, 20-090 Lublin, Poland; wiktorwojczakowski@gmail.com (W.W.); magdalenaszepietowska@wp.pl (M.S.); b.leszczynska@umlub.pl (B.L.-G.); 2Department of Interventional Cardiology and Cardiac Arrhythmias, Medical University of Lodz, 90-549 Łódź, Poland; filipd10@gmail.com; 3Institute of Medical Sciences, State School of Higher Education in Chełm, 22-100 Chełm, Poland; hslodzinski@pwszchelm.edu.pl

**Keywords:** preeclampsia, gestational hypertension, cardiovascular risk, offspring, fetal programming

## Abstract

There is growing evidence of long-term cardiovascular sequelae in children after in utero exposure to preeclampsia. Maternal hypertension and/or placental ischaemia during pregnancy increase the risk of hypertension, stroke, diabetes, and cardiovascular disease (CVD) in the offspring later in life. The mechanisms associated with CVD seem to be a combination of genetic, molecular, and environmental factors which can be defined as fetal and postnatal programming. The aim of this paper is to discuss the relationship between pregnancy complicated by preeclampsia and possibility of CVD in the offspring. Unfortunately, due to its multifactorial nature, a clear dependency mechanism between preeclampsia and CVD is difficult to establish.

## 1. Introduction

According to the World Health Organization (WHO), preeclampsia (PE) is an important problem in obstetrics, affecting 2–8% of pregnancies worldwide [1]. It is one of the most common causes of maternal morbidity and mortality [2].

PE usually presents after 20 weeks of gestation with characteristic signs of hypertension and proteinuria. Clinical symptoms include headache, blurred vision, epigastric pain, nausea, and vomiting. Laboratory investigations show a wide range of abnormalities such as thrombocytopenia, raised serum creatinine, and abnormal liver function tests, mainly raised liver enzymes [3].

Physiological pregnancy is characterized by high blood volume, low vascular resistance, and blood pressure decrease, especially in the first and second trimesters. In PE, on the other hand, low circulating blood volume, high blood pressure, and high vascular resistance are observed [4]. There are many theories of the pathogenesis of PE, but none of them reveals a definitive trigger for the condition. The improper recognition by the maternal immune system of fetal alloantigens deriving from the father (immunological theory), damage to the endothelium, insufficient blood flow through the placenta, increased vascular reactivity, imbalance between prostacyclin and thromboxane production, and decreased glomerular filtration rate with water and sodium retention, are all taken into consideration as possible causes of PE. The common element of these theories is generalized vasospasms [5,6].

Two manifestations of PE, fetal and maternal, have been proposed. The fetal manifestation mainly affects primigravidas with a normal body mass index (BMI). It results from abnormal trophoblast invasion, insufficient placental perfusion and vascular endothelial damage and may present as early as the second trimester. An earlier occurrence of PE is an unfavorable prognostic sign for both the mother and the fetus. This is because fetal form is more often complicated by placental insufficiency. Intrauterine growth restriction (IUGR), hypoxia and intrauterine death, prematurity and placental abruption can occur [5].

The maternal form is more often seen in pregnant women after 34 weeks of gestation with risk factors such as insulin resistance, obesity, diabetes mellitus, chronic hypertension, dyslipidemia, hyperhomocysteinemia, autoimmune diseases, and thrombophilias. Maternal systemic inflammatory response, exacerbated in PE, is responsible for microvascular damage. In this form, clinical symptoms predominate in pregnant woman [7].

The American Heart Association estimates that in 2017, about 17.8 million people died from cardiovascular disease (CVD), which constitutes the primary cause of global mortality. Every year, more people die from CVD than from any other disease [8]. Strategies to prevent CVD are urgently required as the global obesity epidemic grows and will increase the proportion of people at risk of CVD.

The aim of this paper is to discuss the relationship between pregnancy complicated by PE and the possibility of CVD in the offspring.

## 2. Consequences for Offspring

### 2.1. Prematurity

Prematurity is the birth of a baby between 22 and 37 weeks of gestation [9]. Children born prematurely are more likely to die, have worse educational scores, and are at risk of neurological disorders, chronic lung disease, blindness, deafness, and are hospitalized more often. The risk of these adverse outcomes increases as the gestational age at birth decreases [10]. Nearly 10% of the general population is born prematurely [11,12], with PE responsible for 36% of these cases [13]. Babies born from pregnancies complicated by PE are also more likely to suffer from complications of preterm birth.

PE is one of the main iatrogenic causes of preterm labor, as delivery remains the only definitive treatment for this condition [14]. As many as 23% of pregnancies in the United States are finished due to PE [15].

Prematurity is the leading cause of neonatal morbidity and mortality worldwide [16]. Harmon et al. noticed an increased fetal death rate of 5.2 per 1000 among pregnancies with PE versus 3.6 per 1000 in uncomplicated pregnancies [17]. Relative risk of stillbirth was markedly increased with early onset of PE. At 26 weeks of gestation, there were 11.6 stillbirths per 1000 pregnancies with PE, compared to 0.1 stillbirths per 1000 uncomplicated pregnancies. This risk declined with the progress of pregnancy but still remained much higher than in pregnancy without PE [17].

### 2.2. Blood Pressure

There is growing body of evidence of long-term cardiovascular sequelae in mothers following preeclamptic pregnancies [4,18,19]. After in utero exposure, their children also appear to suffer this risk. In 1993, the Barker hypothesis was proposed [20]. Due to this concept, hypertension and/or placental ischaemia during pregnancy increase the risk of hypertension, stroke, diabetes, and CVD in the offspring later in their lives [20,21].

A systematic review performed by Davis et al. showed a 2.39 mm Hg increase in systolic blood pressure (SBP) and a 1.35 mm Hg increase in diastolic blood pressure (DBP) in children and youngsters born to preeclamptic mothers [22]. Davis et al. also analyzed four studies of term-born offspring which presented nearly the same blood pressure increase, with a 2.26 mmHg higher SBP, and a 1.48 mmHg higher DBP [22].

A similar meta-analysis was performed by Andraweera et al., who investigated 15 articles focusing on SBP and 14 on DBP. The offspring of preeclamptic mothers showed higher mean values of both SBP and DBP, with a 5.17 mm Hg and 4.06 mm Hg increase respectively, compared to offspring of non-preeclamptic mothers [23]. These findings confirm a significant increase in mean blood pressure of children born to preeclamptic mothers, though the results for particular studies were not the same for each meta-analysis.

A 2.4 mmHg rise in SBP is connected with increased mortality from ischemic heart disease by 8% and from stroke by 12% [24]. Moreover, youngsters with in utero exposure have 2.5 times higher risk to have scores above the 75th centile of global lifetime risk (QRISK) [25]. Thirty percent of 20-year-old adults with high blood pressure were born from mothers who suffered from pregnancy-induced hypertension. Another study indicated that they were more likely to take antihypertensive medications before 50 years of age [26].

Lazdam et al. observed that early-onset PE was associated with higher risk of hypertension in mothers and their offspring [27]. The authors carried out a 60 year follow-up of the Helsinki birth cohort that showed a 1.1–1.5 increased relative risk of hypertension in the offspring [27]. Data collected at 6 to 13 years after pregnancy showed higher SBP in these children compared to children born from mothers with late-onset PE [27]. These findings agree with those of another study which found a 6 mmHg increase in peripheral and central SBP in children born from mothers with early-onset PE [22].

PE with vasculotoxic factors that cross the placenta causes defects in the systemic and pulmonary circulation of the offspring. This leads to exaggerated hypoxic pulmonary hypertension, and may be responsible for premature CVD in the systemic circulation later in life [28]. A comprehensive British cohort of maternal-offspring pairs included 3537 mothers and their 4654 children [29]. This study found an association between both gestational hypertension and PE, and higher values of blood pressure in offspring. However, the study did not confirm any associations with flow-mediated dilatation, radial pulse wave velocity, brachial distensibility coefficient, lipids, apolipoproteins, or inflammatory markers [29].

### 2.3. Body Mass Index and Lipids

A meta-analysis of 8 studies with 39,611 children, adolescents, and young adults showed a significant increase of 0.57 kg in the offspring of preeclamptic women. The assessment of term-born children, including only newborns with a weight greater than 2.5 kg, also presented a significant increase in BMI [22].

A Turkish study assessed 60 neonates born by mothers with PE. The authors observed significantly higher aortic intima-media thickness (aIMT) measurements and serum triglyceride levels [30]. Concentrations of serum HDL were significantly lower in the group of PE children. The authors confirmed that children born to mothers with PE had significantly higher aIMT with lipid alterations. This may play a role in the pathogenesis of atherosclerosis in adult life [30].

Andraweera et al., in their evaluation of risk factors for increased CVD in children exposed to PE in utero, found 0.36 kg/m^2^ greater BMI in these children [23].

In contrast to increased BMI in later life, neonates born from pregnancies complicated by PE have significantly lower birth weights. The study by Odegard et al. revealed a 5% reduction in birth weight among newborns after in utero exposure to PE. This reduction was greater in cases of severe disease and even 23% with early onset [31]. The general risk of being small for gestational age (SGA) was four times higher in infants born from PE pregnancies than from the controls [31].

Little data is available regarding cholesterol levels in offspring. Kvehaugen et al. assessed endothelial function and circulating biomarkers in women and their children after PE [32]. They found significantly higher median serum concentrations of total cholesterol in children exposed to PE when compared to children from uncomplicated pregnancies [32]. Andraweera et al. did not confirm the above findings. The authors described no differences in total cholesterol, triglycerides, LDL, or HDL levels between offspring from PE and uncomplicated pregnancies, regardless of the use of cord blood [23].

### 2.4. Congenital Heart Disease

Congenital heart disease (CHD) is a major cause of infant morbidity and mortality, and also the most common birth defect [33,34,35,36,37]. CHD is diagnosed in up to 3% of all children and constitutes about 28% of all major congenital anomalies [34,37]. The possible connection between CHD and PE has been examined. A study performed in Nigeria found CHD in 21.2% newborns from women with PE and in 3.3% newborns from healthy mothers by the end of the 4th week of life [38]. Furthermore, isolated atrial and ventricular septal defects were observed in 4.4% of the offspring of women with PE [38]. A similar conclusion was postulated by Auger et al., from Canada [39]. The authors confirmed the association between preterm birth and PE/eclampsia (24.1%) as well. A large cohort study was performed in Denmark from 1978 to 2011 [40]. It included almost 2 million pregnancies without chromosomal abnormalities lasting at least 20 weeks of gestation. In this study, the risk of CHD was increased 7-fold in offspring from pregnancies with early-onset PE and 3-fold in offspring from pregnancies with late-onset disease [40].

### 2.5. Long-Term Cardiovascular Morbidity

Long-term cardiovascular effects in offspring were studied by researchers from Israel in a cohort of 231,298 deliveries, with 3.2% pregnancies complicated by mild PE and 0.9% by severe PE [41]. The CVDs recorded in this study included cardiomyopathy (n = 38), hypertension (n = 153), pulmonary heart disease (n = 32), arrhythmias (n = 303), heart failure (n = 84), and other CVDs (n = 559). The authors observed a linear relationship between CVD and PE, with the risk increasing with the severity of PE [41]. This study was the first to demonstrate an association between PE and arrhythmias or heart failure. However, it did not confirm such a relationship between PE and either cardiomyopathy or pulmonary heart disease [41]. Importantly, severe PE was found to be an independent risk factor for long-term cardiovascular morbidity only in term-born children [41]. This stands in contrast to previous studies which have shown a higher frequency of cardiovascular morbidity in offspring from pregnancies complicated by the early onset of severe PE before 34 weeks of gestation [27,42].

### 2.6. Cardiac Structure and Function

Most papers focus on hypertension in offspring from pregnancies complicated by PE, whilst a few deal with changes in the heart itself. A study by Timpka et al. using echocardiography in 1592 adolescent subjects reported a concentric type of remodeling, which means increased relative wall thickness (RWT) and reduced left ventricular end-diastolic volume (LVED) [43]. The above findings were confirmed by another study, although it targeted the offspring of women with hypertension, not PE. By 3 months of age, children had a significantly greater left (LVMI) and right ventricular mass index (RVMI) as well as smaller RVED [44]. It is worth noting that real-time 3-dimensional echocardiographic assessment in the second and third trimesters revealed a larger right ventricle (RV) than left ventricle (LV) in normal fetuses compared to CHD fetuses, but without any differences between the LV and RV in mass, stroke volume, cardiac output, and combined cardiac output [45]. As both studies have shown changes in cardiac structure, there is no evidence of cardiac dysfunction [43,46].

Diastolic dysfunction has been reported in two other studies. Left ventricle diastolic dysfunction (LVDD) was detected in premature infants born to preeclamptic mothers in the first week after delivery [47]. The older group of children, aged 5–8, was found to have smaller hearts, increased heart rates, and increased late diastolic velocity at the mitral valve attachments [48]. The study of Zhou et al. seemed to confirm that there is diastolic impairment of LV in fetuses during PE pregnancy with or without IUGR. This was even more pronounced with preterm delivery at less than 34 gestational weeks or stillbirth [49].

However, the study performed by Hoodboy et al. did not find any differences between groups of children exposed and unexposed to PE, assessing over 20 parameters including cardiac morphometry, systolic function, diastolic function, and timing assessment [50]. No significant associations were observed between groups and cardiac parameters, both in function and structure, including basal septal hypertrophy, basal septal longitudinal strain, LV mass, and LV mass Z score. Their results seem to be inconsistent with existing data concerning changes in cardiac structure and function in children exposed to PE.

It is worth considering why such changes occur in children’s hearts and what consequences this may have in the future. It can be stated that a poorly perfused placenta could lead to oxidative stress within fetal tissues and in the placenta itself. Moreover, the developing heart of the affected child has to deal with increased impedance because of poorly constructed placental microcirculation [50]. The other issue is the development of the heart itself. Over the first and second trimesters, intensive replication of cardiomyocytes occurs, resulting in heart growth with concomitant increase in microcirculation. The third trimester is the period of maturation of the cardiomyocytes, i.e., replication of the nuclear genome in the absence of mitosis, leading to increased nuclear gene content and polyploidy, known as endoreduplication [51]. By the time of birth, 70% to 80% of these cells are terminally differentiated. They are no longer proliferating, practically completing the division process within a few months [52]. Of course, the heart continues to develop after this period, but through cardiomyocyte hypertrophy. Fetal myocardial maturation is influenced by two factors: hormone-mediated regulation and, more interestingly, the hemodynamic load [53]. Increased systolic load leads to a short spurt of proliferation among fetal cardiomyocytes and the following cessation of proliferation [54]. Instead, there is a steady increase in terminal differentiation [54]. A reduced number of cardiomyocytes and their accelerated maturation leads to abnormal hypertrophy, resulting in altered heart chamber anatomical characteristics [54]. These experimental data were obtained in large mammals [54]. It can be conditionally assumed that similar processes occur in human fetuses, and the early onset of PE may further adversely affect heart formation.

### 2.7. New Risk Factors

Well-known CVD risk factors include elevated blood pressure, increased BMI, and elevated levels of circulating lipoproteins. Augmentation index (Aix), microvascular function, and suprasystolic pulse pressure (ssPP) are indicated as novel risk factors. Aix is a parameter connected with vascular stiffness. Many studies suggest an association between Aix and CVD risk and higher mortality rate [55,56,57].

The impairment of microvascular function occurs long before the onset of clinical symptoms [58]. This function may be assessed using the two parameters of peak perfusion: time to max (TM), and recovery time (time to half, TH2) using laser Doppler perfusion monitoring [59]. With regard to ssPP, this risk factor is associated with obesity in children. This parameter is a non-invasive measure of vascular stiffness [60].

Plummer et al. focused on the hemodynamic profiles of children aged 8-10 years exposed to PE or gestational hypertension in utero [59]. Children exposed to PE had significantly increased Aix, ssPP, TM, and TH2. There was no difference in peak perfusion. This study defined these parameters according to the childrens’ gender. Female offspring delivered by preeclamptic mothers had significantly decreased Aix and ssPP, but increased TM and TH2 when compared to male children. Males had increased vascular stiffness, whereas females were characterized by endothelial dysfunction [59].

The increased values of Aix and ssPP, as well as an impairment of microcirculation observed in children born from pregnancies with PE, are probably connected with the fact that larger vessels may be less compliant [59]. In spite of increasing TM, there was no difference in peak perfusion. According to Plummer et al. this observation may suggest an impaired post-ischaemic vasodilation and delay in the endothelial independent myogenic response [59]. The authors did not observe a decrease in peak perfusion associated with the increased recovery, but did observe that endothelial function was compensating for the endothelial independent pathway and the beginnings of vascular stiffness in the bigger vessels [59]. One of the most important findings in this study was the fact that microvascular impairment progressed more slowly in the cases of children delivered by mothers with gestational hypertension compared to those from pregnancies complicated by PE [59]. Nevertheless, the authors did not observe any differences between children born from uncomplicated pregnancies and pregnancies complicated by PE or gestational hypertension, nor male between female children. Only the recovery time was decreased in the case of females [59].

The carotid artery intima-media thickness (cIMT) is an essential parameter of CVD [60], which is measured in B-mode ultrasound examination of the carotid tree as a typical double line of the arterial wall [61]. This marker of subclinical atherosclerosis is an estimated valid tool to assess cardiovascular risk in adults. It could also be a useful parameter for such assessment in children [62]. Lazdam et al. showed that offspring from hypertensive pregnancies had significantly greater cIMT [63]. However, it should be emphasized that prematurity, which is usually connected to PE pregnancies, is an independent risk factor for increased cIMT [64]. cIMT varies with age, sex, and race in adults, and as such, establishing the causality of PE and vascular remodeling may be difficult [62].

### 2.8. Kidneys

A kidney dysfunction represents another important aspect in the analysis of cardiovascular risks in preeclamptic offspring. The renal impact on the blood pressure is well-defined. PE often coexists with low birth weight and prematurity, which in turn is associated with the reduced number of nephrons and cardiomyocytes [65]. This reduction in the number of nephrons decreases the rate of renal filtration. Consequently, more blood circulates in the bloodstream, and this translates into higher pressure [66]. There is also an issue of the glomerular hypertrophy and reduced renal vascular dilation, which may contribute to hypertension [67,68].

Many of the proposed mechanisms of hypertension observed in these offspring are due to the function and development of the kidneys [69]. Since human nephrogenesis occurs in the third trimester of pregnancy, when PE is often the most staggering, it is reasonable that the resulting changes in blood flow and circulatory factors may adversely affect the fetal kidneys. PE is known to be the major cause of IUGR and preterm labor [70,71]. IUGR, due to uteroplacental insufficiency, results in a nephron deficiency and glomerular hypertrophy in both male and female rats [72]. Human studies appear to support these data because infants with SGA, which is often associated with placental insufficiency, have an increased risk of end-stage kidney disease compared to controls [73].

Physiologically, fetal renal arteries are high-resistance vessels. After birth, there is a rapid decrease in their resistance, with a simultaneous increase in blood flow through the kidneys [74]. Moreover, the glomerular filtration rate in the fetal kidneys begins to increase rapidly from 34 weeks of gestation [75]. It is associated with an increase in urinary flow rate in the fetus [74]. During early renal development, the renin-angiotensin system is upregulated, which leads to the vasoconstriction of the renal arteries [76]. Fetal serum renin concentrations are consistently higher than those of the mother. There is also an increase in sympathetic tone in the renal vessels, as well as an increased sensitivity to adenosine in the fetus compared to that of the neonate, which results in an increase in the tone of the renal arteries. These vasoconstricting effects are counterbalanced by vasodilatory peptides including prostaglandins, nitric oxide (NO) and the Kallikrein–Kinin system [74]. Meanwhile, placental hypoxia in PE may result in dysfunction of the maternal vascular endothelium and the disturbance of the balance of vasoactive compounds [77,78]. This vasoactive imbalance includes the increased formation of vasoconstrictors (such as thromboxane A2 [TxA2] and endothelin), decreased formation of vasodilators (including prostacyclin [PC] and NO), and increased vascular sensitivity to angiotensin II [79,80]. It seems that levels of TxA2 and PC in the fetal circulation increase during the course of PE [81]. Levels of TxA2 and PC are measured indirectly using the levels of their stable metabolites thromboxane B2 (TXB2) and 6-keto-prostaglandin F1α (6KPG), respectively. The ratio of TXB2 / 6KPG in fetal umbilical cord blood in PE is lower compared to normal pregnancy, which suggests an increased relative concentration of PC [74,81]. In the adult population, it has been shown that abnormalities in renal artery resistance are associated with the progression of renal dysfunction in patients with chronic kidney disease [82]. Moreover, placental insufficiency in PE is associated with a reduction in the number of nephrons in the offspring [83], which may affect their long-term health.

## 3. Pharmacotherapy

A meta-regression analysis of randomized clinical trials by Magee et al. assessed 42 trials in 3892 women. The authors revealed that antihypertensive therapy could increase the risk of IUGR [84]. They also described a significant relationship between the antihypertensive-induced decrease in mean arterial pressure and the risk of SGA or lower birth weight. Heida et al. showed an association between intrauterine labetalol exposure and hypotension in neonates. They observed this effect in 29.1% of children exposed to labetalol in utero vs. 7.4% in controls [85]. More research is needed into the effects of the treatment used in the mothers with PE.

## 4. Molecular Aspects

A relationship between CVD in the offspring and pregnancies complicated by PE seems to be a combination of genetic, molecular, and environmental factors that can be defined as fetal and postnatal programming. The primary mechanism for the onset of chronic disease may be altered by changes in the homeostatic set points, including the hypothalamic–pituitary–adrenal axis, vascular structure and sensitivity, the rennin–angiotensin–aldosterone system (RAAS), and metabolic and hormonal aspects [86,87,88,89]. It is well-known that vascular and endothelial dysfunctions play a key role in the development and progression of PE [90]. Rodent studies showed that vascular function was altered in the offspring of mice with soluble fms-like tyrosine kinase-1 (sFlt-1) [87]. An anti-angiogenic status was found in the adulthood of PE offspring with elevated plasma levels of sFlt-1 and soluble endoglin associated with elevated blood pressure [91].

The kidneys can be programmed by a variety of perinatal insults, including placental insufficiency [92]. Singh et al. observed that reduced excretory capacity may be due to impaired expression of renal sodium transporters and channels [93]. In addition, RAAS and sympathetic nervous system programming may also be connected to hypertension in the offspring. Placental insufficiency has been reported to affect RAAS programming [94,95]. Animal models have revealed increased sensitivity to angiotensin II [96]. Furthermore, the blockade of RAAS by the angiotensin-converting enzyme inhibitor (ACEI) or angiotensin II type 1 receptor blockers may prevent the development of hypertension in the adult offspring of female rats with reduced uterine perfusion pressure (RUPP) [97,98]. It could also confirm the involvement of RAAS in the fetal programming.

Prenatal exposure to elevated testosterone levels in mothers with PE is associated with increased blood pressure in the female offspring during adulthood [89,99,100]. Animal studies have revealed that elevated androgen concentrations during pregnancy may lead to hyperactivity of the hypothalamic–pituitary–gonadal axis and changes in the expression of steroid genes in the gonads of the offspring, resulting in increased testosterone production [101,102]. More et al. observed that prenatal exposure to elevated testosterone concentrations was associated with a decrease in CYP11B2 expression, leading to a reduction in the plasma aldosterone levels in the offspring, but the plasma volume and balance between sodium and potassium ions were normal [103]. However, the plasma concentrations of vasopressin and angiotensin II, the vascular response to angiotensin II, and blood pressure were all increased in female offspring exposed to higher testosterone levels, which may serve as a compensatory response to maintain plasma volume and the sodium and potassium balances. Henley et al. demonstrated that levels of adrenocorticotropic hormone and cortisol were significantly increased in the 17-year-old offspring of women with PE, which is suggestive of the reprogramming of the hypothalamic–pituitary–adrenal axis [86].

Models of IUGR and maternal protein restriction have shown alterations of RAAS in offspring subjected to these insults in utero [69]. Woods et. al. used the maternal protein restriction model in order to show that the intrarenal RAAS is suppressed in the male offspring of these pregnancies, and to propose a link between the reduction of developed glomeruli and later hypertension [104]. They also observed the relative protection of the female offspring subjected to the same gestational insult [105]. Hypersensitivity of blood pressure to angiotensin II has also been demonstrated in the offspring of the RUPP-IUGR model in a similarly sex-specific fashion [69,106].

Nitric oxide is supposed to contribute to the epithelial–mesenchymal transformation in the endocardial cushion areas, myocardial survival and angiogenesis, and myocardial remodeling. Impaired NO production in the heart causes structural defects that lead to heart failure and increased mortality. It is known that the inhibition of NO during cardiac development promotes bicuspid aortic valve defects, congenital septal defects, and increased cardiomyocyte apoptosis [69].

The vasodilatory response of normal pregnancy is highly dependent upon NO, with the levels of both NO and NO synthase (NOS) shown to be consistently increased in both animals and humans with normal pregnancies [69]. NO, by diffusing into the vascular smooth muscle, binding guanylyl cyclase, and producing the second messenger molecule cyclic guanosine monophosphate (cGMP), activates protein kinase G to reduce the calcium concentration and cause vasodilation [107]. The endothelium is also a major source of NO, from the NOS isoform called endothelial NOS (NOS3) [108]. NO production, as measured by its metabolic end products (NOx) and NOS3 expression, seems to be decreased in the animal models of PE, while in humans NO production is likely to be reduced only in the kidney [69,109,110].

## 5. Genetics

### 5.1. Hereditary Factors

The hypotheses of both Barker and the Developmental Origins of Health and Disease are based on the concept that exposure to negative conditions during fetal life at a critical time of organ development may increase risk of disease in later life [20,111,112]. These theories may be also emphatically confirmed in the case of PE. The pathogenesis of PE includes excessive inflammation, ischemia/perfusion, angiogenic imbalances, and disturbances in the renin angiotensin system [113,114].

In PE, the placenta releases factors such as sFlt and sEng into the maternal circulation due to stress caused by ischemia/hypoxia [115,116]. This usually results in the formation of harmful reactive oxygen species, inflammation, and lipid peroxidation [117,118,119,120,121,122]. A study using a mouse model mimicking placental PE showed that overexpression of sFlt causes elevated SBP and DBP, but only in male offspring [123]. Researchers in Turkey indicated that oxidative stress and damage of DNA were present in the cord blood of children born from PE pregnancies [124], which is well-known that it causes a reduction in fetal growth [125]. Another experimental study conducted on rats showed inflammation and atherosclerosis in offspring exposed to hypoxia [126].

Hu et al. found changes in the immune system, with fewer T regulatory cells (Treg) persisting into early childhood [127]. Taking into account the fact that inflammation plays a significant role in the pathogenesis of atherosclerosis and CVD, this process of immune-programming in PE pregnancies may lead to CVD [128,129,130,131].

Advances in genetic technology have made a major impact on clinical practices [132]. Clinical genome and exome sequencing are crucial tools in the implementation of prognostic and individualized medicine [133]. There is a growing body of research looking for links between genetics and PE [134]. A study from 2004 examined Swedish families with gestational hypertension and PE [135]. Mothers–daughters and full sisters shared more similarities than half-sisters regarding PE and gestational hypertension. The authors concluded that heritability estimated for PE in 31%, for gestational hypertension in 20%, and for pregnancy induced hypertension in 28% of cases [135].

The incidence of PE is higher in mothers who were born from pregnancies complicated by PE, in fathers who previously conceived a pregnancy complicated by PE with another partner [136], and in fathers who themselves were born from a pregnancy complicated by PE [137]. A study examined the genetic dissection locus on chromosome 2q22, more specifically single nucleotide polymorphisms (SNPs) [138]. Johnson et al. identified four independent SNPs to be significantly associated with PE. Furthermore, these SNPs exhibited evidence of pleiotropy with several quantitative CVD-related traits, indicating that these two diseases share underlying genetic mechanisms [138].

Another study analyzed the connection between SNPs in the insulin receptor gene (INSR) and PE. In this study, the prevalence of the INSR AA genotype was significantly higher in preeclamptic women when compared to those with uncomplicated pregnancies [139].

A study performed in the Mexican population also confirmed these findings [140]. Australian researchers found a relationship between SNPs and CVD risk factors, such as body weight, blood triglycerides, and glucose levels [141]. In the future, there is a hope for eliminating such SNPs by turning off their expression using genetic engineering techniques.

Modern methods were also applied in a study investigating plasma angiopoietin 1 (ANG-1) concentrations. The TT genotype of the angiopoietin 1 gene (ANGPT1) was linked with increased plasma ANG-1 levels compared to the AA genotype. The study determined that the prevalence of the TT genotype was significantly decreased in women with PE. This indicates that certain genotypes may play a protective role against pregnancy complications, such as PE, hypertensive SGA, and the abnormal uterine artery Doppler indices [142].

Another preliminary study demonstrated that the homozygosity of the CC allele of the KDR-604T/C polymorphism in both the father and the infant is associated with higher risk of PE and SGA [137]. KDR-604 is the main receptor for vascular endothelial growth factor A, which regulates the development of the placental vasculature [137].

Inherited polymorphisms, epigenetic factors, and altered microRNA expression play a crucial role in the development of a normal pregnancy, as well as in pregnancies complicated by PE [143,144,145]. Maternal levels of sFlt-1 were correlated with human umbilical vein endothelial cells (HUVECs). HUVECs from affected offspring had lower vasculogenic capacity, which was associated with reduction in microvascular density in the early postnatal period [144]. This highlights the role of genetic factors in the pathogenesis of PE. Yu et al. found that the upregulation of miR-146a expression in HUVECs was associated with reduced postnatal vasculogenesis. They also confirmed that miR-146a levels may be a good predictor of microvascular development during the first three months of life [145].

### 5.2. Epigenetics

Epigenetics refers to changes in gene expression that are not related to the nucleotide sequence [146]. Epigenetics deals with processes such as DNA methylation, non-coding RNA (ncRNA) particles, and modifications of histones. There is a growing evidence of the decisive influence of environmental and lifestyle factors on gene expression and disease development. Moreover, epigenetic modifications are used as diagnostic and prognostic biomarkers. The methylation of DNA is responsible for many fundamental molecular phenomena, such as imprinting and inactivation of the X-chromosome, repression of transposable elements, aging, and carcinogenesis [146]. It is well known that gene repression is associated with the hypermethylation of CpG islands and the regulatory regions of promoters. Histone modifications affect chromatin packing and the disponibility of the transcription factors to the regulatory DNA sequences [147]. In turn, the members of non-coding RNA (ncRNA) family influence protein transcription, translation, trafficking, and folding [146].

The work studying children born SGA very prematurely or following in vitro fertilization suggested that epigenetics was a potential explanation for environmentally induced changes in gene expression in offspring [148].

Several important papers have been published on the role of epigenetics in placental gene modification. Hogg et al. examined the association between early-onset PE and the altered methylation of cortisol-signaling genes and steroidogenic genes in the placenta [149]. Cortisol was present in the placentas of preeclamptic women but was not in those from normal pregnancies [149,150,151,152]. All these data showed that dysregulation of the placental epigenome plays an important role in an early onset PE [149]. Blair et al. examined 20 chorionic villi samples from early onset PE placentas and 20 gestational age-matched controls from preterm births [153]. They identified 38,840 CpG sites with significant alterations in DNA methylation patterns in early onset PE [153]. Interestingly, there are a number of studies that identified different methylation patterns in placentas from both normal and complicated pregnancies. These genes include those involved in trophoblast proliferation, differentiation, and invasion [154,155].

Epigenetics gives a potential explanation for the mechanism linking PE to CVD in offspring and its further transmission to subsequent generations [156,157]. Moreover, when assessed in the placenta or cord blood, changes in DNA methylation may be potential biomarkers of exposure to PE. This is because these changes occur not only in the placenta but are also found in fetal cells [154,155,158].

The role of genetics and epigenetics has been convincingly confirmed in the discussion above. Other influences besides genetic factors, should be taken into account, considering the origins of PE. Jayet el al. observed that offspring exposed to PE showed significant pulmonary and systemic vascular dysfunction [28]. This stands in contrast to a study from Norway by Alsnes et al. in which siblings had nearly identical risk factors for hypertension, although they were born after a normotensive pregnancy [159]. This study included 210 siblings, which is significantly more than in the previously mentioned paper. Another important issue is the severity of PE and the duration of fetal exposure to PE. Longer time from diagnosis of PE to delivery was associated with increased long-term morbidity in offspring [160].

Environmental and lifestyle factors such as diet, sedentary lifestyle, lack of exercise, and low socio-economic status play an important role in the morbidity of parents and their children. These outcomes can be connected following the same patterns of lifestyle running in those families. The aforementioned factors could also be responsible for morbidity from CVD in these children. Genetic factors alone can be liable for developing the disease in adulthood. The theory of the second hit states that other factors are also required to unmask “programmed” hereditary impairments. Pregnancy itself can be considered a second hit in the pathogenetic chain leading to PE, and then CVD in the further life of both the mother and her offspring [161,162].

## 6. Conclusions

The analysis of selected scientific reports shows that there is a correlation between PE and its effect on the long-term development of cardiovascular disorders. The evidence presented supports the view that this correlation negatively affects, among other things, cardiovascular morbidity, blood pressure, and BMI. It also affects the development of congenital heart disease and kidney disorders, which further increases the risk of CVD development. However, due to inconsistencies in the analyzed studies, there are still factors, such as lipids or the structure and function of the heart itself, that require in-depth research.

It is difficult to establish a clear causal mechanism between PE and CVD, due to its multifactorial nature. The role of genetics and epigenetics has been confirmed. Nevertheless, it is impossible to accurately determine the severity and type of inherited susceptibility to the disease. It should be remembered that the pharmacotherapy used during pregnancy carries the risk of adverse effects on the fetus.

Finally, environmental aspects, unhealthy lifestyle, and low socio-economic status overlap with other risk factors for PE, thereby distorting the assessment of correlation with CVD.

Therefore, in order to develop effective preventive strategies for this target group, i.e., both the screening tests model and the standards of treatment, further studies of the effect of PE on the cardiovascular risk in children are essential.

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
