# Peer review of "Preeclampsia and Cardiovascular Risk for Offspring"

_jcm, 2021, doi:10.3390/jcm10143154_

Round 1

Reviewer 1 Report

I am please to review this extensive literature review of Pre-eclampsia and the cardiovascular risk for offspring. 

The first few sections are much more successful for the reader than the last sections. I am unsure of the word limit for these sorts of articles but overall I would say it could be edited down quite significantly.  I found section 5 long and very complicated being unfamiliar with many of the specifics of for example single nucleotide polymorphisms and I think this could be much shorter.

specific comments

line 127

the first paragraph- it is not clear when the BMI was measured- infants, children, adolescents- please clarify

line 187/188 Don't understand - what is a concentric type of modelling?

line 332/333- rephrase this sentence

Are you trying to say more research is needed into the effects of treatment used in the mothers with PE?

line 355- what is 'dams of the adult offspring' ?

line 431/432 Don't understand this sentence - please rephrase

line 455 what is CRISPS - I don't see this defined anywhere

line 562 /563 this sentence needs rephrased to make sense

There  are a few minor grammar and spelling issues

ne 506 fetal/fetus instead of foetal/foetal

consistency with spelling e.g. dysfunction not disfunction

line 116 cause defects

line 131 A Turkish study

line 132 higher aortic ...

line 149 confirm

line 209 including basal septal hypertrophy

line 292 take out 'On the other hand ' just start 'PE is known to be ...'

line 358 take out 'Another aspect...' just start 'Prenatal exposure ....'

line 374 take out 'Moreover...' just start 'Models of ..'

line 403 development may increase risk of 

line 407 In PE the placenta .....into the maternal circulation ...'

line 503 first two sentences need rephrasing e.g. 'Several important papers have been published on the role etc . One examined ...'

the diagram page 13 added nothing at all to the paper

Author Response

Dear Reviewer,

Thank you very much for your valuable and thoughtful comments. We find all your remarks spot on therefore we have made a point-by-point correction of the manuscript according to your suggestions. We have rearranged our manuscript. We have reduced the section entitled “Genetics”.

specific comments

line 127

the first paragraph- it is not clear when the BMI was measured- infants, children, adolescents- please clarify

Following your advice, we have corrected:

“A meta-analysis of 8 studies with 39611 participants children, adolescents and young adults showed a significant increase of 0.57 kg in offspring of preeclamptic women.”

line 187/188 Don't understand - what is a concentric type of modelling?

A concentric type of remodeling is one of types of ventricular remodeling. It generally exhibits a trend toward higher left ventricular (LV) mass than is seen with a truly normal geometry; it appears to be an early response to a LV pressure overload.

We have rearranged this sentence as follows:

“A study of Timka et al. using echocardiography in 1592 adolescent subjects reported a concentric type of remodeling, which means the increased relative wall thickness (RWT) and reduced left ventricular enddiastolic volume (LVED) [44].”

line 332/333- rephrase this sentence

Are you trying to say more research is needed into the effects of treatment used in the mothers with PE?

Following your advice, we have corrected:

“More research is needed into the effects of treatment used in the mothers with PE.”

line 355- what is 'dams of the adult offspring ?

We have replaced “dams” by “female rats”

line 431/432 Don't understand this sentence - please rephrase

The sentence was removed.

line 455 what is CRISPS - I don't see this defined anywhere

The word “CRISPS” was removed.

line 562 /563 this sentence needs rephrased to make sense

Following your advice, we have rephrased:

“The role of genetics, inheritance and epigenetics has been convincingly confirmed. above, yet,  it cannot be responsibly isolated for the severity and type of inherited susceptibility to disease. Nevertheless, it is impossible to accurately determine the severity and type of inherited susceptibility to the disease.”

There  are a few minor grammar and spelling issues

ne 506 fetal/fetus instead of foetal/foetal

We have replaced “foetal/foetus” by “fetal/fetus”

consistency with spelling e.g. dysfunction not disfunction

The sentence with the word “disfunction” was removed.

line 116 cause defects

It was corrected.

line 131 A Turkish study

It was corrected.

line 132 higher aortic ...

It was corrected.

line 149 confirm

It was corrected.

line 209 including basal septal hypertrophy

It was corrected.

line 292 take out 'On the other hand ' just start 'PE is known to be ...'

It was corrected.

line 358 take out 'Another aspect...' just start 'Prenatal exposure ....'

It was corrected.

line 374 take out 'Moreover...' just start 'Models of ..'

It was corrected.

line 403 development may increase risk of 

It was corrected.

line 407 In PE the placenta .....into the maternal circulation ...'

It was corrected.

line 503 first two sentences need rephrasing e.g. 'Several important papers have been published on the role etc . One examined ...'

It was corrected.

the diagram page 13 added nothing at all to the paper

We have decided to delete Figure 1 from the text of our manuscript. This figure could be a graphical abstract.

             We would like to take this opportunity to thank the Reviewers and Editors for all the valuable and highly perceptive remarks which have definitely made a substantial contribution to the quality of our paper. Some English language issues have been corrected.

            Thank you for considering our manuscript for publication. We appreciate your time and look forward to hearing from you.

Yours faithfully,

Assoc. Prof. Zaneta Kimber-Trojnar, M.D., Ph.D.

Chair and Department of Obstetrics and Perinatology

Medical University of Lublin

Jaczewskiego 8, 20-090 Lublin, Poland

Tel: +48 81 7244 769;

Fax: +48 81 7244 841

Reviewer 2 Report

Dear Mr Editor in chief,
Thank you for asking me to review the  manuscript Manuscript ID: jcm-1260128 entitled “CPreeclampsia and cardiovascular risk for offspring”, which was submitted to the International Journal of Obstetrics & Gynecology,
https://www.mdpi.com/journal/jcm/sections/Obstetrics_Gynecology.

After reading the manuscript,I strongly believe that this paper met the basic criteria to be considered as well written. I would like to recommend it for publication .

Author Response

Dear Reviewer,

Thank you very much for finding the time to read our manuscript. Thank you for considering our manuscript.  

Yours faithfully,

Assoc. Prof. Zaneta Kimber-Trojnar, M.D., Ph.D.

Chair and Department of Obstetrics and Perinatology

Medical University of Lublin

Jaczewskiego 8, 20-090 Lublin, Poland

Tel: +48 81 7244 769;

Fax: +48 81 7244 841
